# Microglial metabolism is a pivotal factor in sexual dimorphism in Alzheimer's disease

Marie-Victoire Guillot-Sestier [1,6], Ana Rubio Araiz[1,6], Virginia Mela[1,5,6], Aline Sayd Gaban[1,6], Eoin O'Neill [1,6], Lisha Joshi[2], Edward T. Chouchani [3,4], Evanna L. Mills [3,4] & Marina A. Lynch [1✉]

Age and sex are major risk factors in Alzheimer's disease (AD) with a higher incidence of the disease in females. Neuroinflammation, which is a hallmark of AD, contributes to disease pathogenesis and is inexorably linked with inappropriate microglial activation and neurodegeneration. We investigated sex-related differences in microglia in APP/PS1 mice and in post-mortem tissue from AD patients. Changes in genes that are indicative of microglial activation were preferentially increased in cells from female APP/PS1 mice and cells from males and females were morphological, metabolically and functionally distinct. Microglia from female APP/PS1 mice were glycolytic and less phagocytic and associated with increased amyloidosis whereas microglia from males were amoeboid and this was also the case in post-mortem tissue from male AD patients, where plaque load was reduced. We propose that the sex-related differences in microglia are likely to explain, at least in part, the sexual dimorphism in AD.

[1] Trinity College Institute for Neuroscience, Trinity College, Dublin 2, Ireland. [2] Gottfried Schatz Research Centre, Medical University of Graz, Graz, Austria. [3] Department of Cancer Biology, Dana–Farber Cancer Institute, Boston, MA, USA. [4] Department of Cell Biology, Harvard Medical School, Boston, MA, USA. [5] Present address: Department of Endocrinology and Nutrition, Instituto de Investigación Biomédica de Malaga (IBIMA), Virgen de la Victoria University Hospital, Málaga University, Malaga, Spain. [6] These authors contributed equally: Marie-Victoire Guillot-Sestier, Ana Rubio Araiz, Virginia Mela, Aline Sayd Gaban, Eoin O'Neill. ✉email: lynchma@tcd.ie

Microglia are particularly dynamic and plastic cells, rapidly adapting to a multitude of stimuli to ensure that neurons are protected and optimally functional. This key neuroprotective role of microglia is amply demonstrated by the impact of dysfunctional microglia on neurons. Transcriptomic analysis of isolated microglia has led to the identification of a profile of homoeostatic microglia that express genes including *Tmem119, P2ry12* and *Cx3cr1*, and microglial phenotypes that are associated with age and neurodegeneration in which there is a decrease in expression of homeostatic genes and an increase in expression of genes that reflect neuroinflammation/upregulated immune function including *Spp1, Itgax, Axl, Lgals3, Clec7a Trem2, Tyrobp* and *Cd68*[1–4]. Specific populations of microglia, described as disease-associated microglia (DAMs) and activated response microglia (ARM) have been identified in 2 models of Alzheimer's disease (AD), 5XFAD mice[5] and App[NL-G-F] mice[6]. However, there have been no systematic studies of sex-related differences in microglia in mouse models of AD, or in AD.

Consistent with data from transcriptomic studies, several groups have reported that microglia in models of AD adopt an activated and inflammatory phenotype. Such cells shift their metabolism towards glycolysis[7] and therefore microglia isolated from the brains of aged mice and APP/PS1 mice are characterised by an inflammatory and glycolytic signature[8,9]. The extent to which the signature of these microglia is sex-dependent is not known.

Here we sought to investigate sex-related changes in microglia and show that the shift in microglial metabolism towards glycolysis in cells from 18 month-old APP/PS1 mice, which is accompanied by inflammatory and phenotypic changes, is more profound in cells from female, compared with male, mice. One consequence of these changes is a deterioration in phagocytic function. Importantly, sex-related differences in microglial phenotype were also identified in post-mortem samples from AD patients and this was associated with increased amyloid load in females, compared with males. This sexual dimorphism in microglia may provide some explanation for the well-described sex-related differences in the clinical manifestation and progression of AD.

## Results

### Sex impacts on differential expression of microglial markers in APP/PS1 and WT mice.
We used multiplexed gene expression analysis (NanoString nCounter) to investigate genotype-related changes in isolated microglia prepared from the brains of male and female mice. The heat maps show marked changes in several genes in cells from APP/PS1, compared with WT, mice with obvious sex-related differences. Specifically, genes that define DAMs and/or ARMs (Fig. 1a)[5,6] and genes that are upregulated in inflammatory conditions like ALS and injury (Fig. 1b) were changed in a genotype- and sex-related manner and, in particular, were markedly increased in microglia from female APP/PS1 mice. These include genes that are linked with altered risk of AD-like *Apoe, Bin1, Trem2* and *Cd33*[10]. Sex-related differences in genes that identify homoeostatic microglia were also observed with marked downregulation of *P2ry12* and upregulation of several genes that have been shown to increase in models of disease like *Ctsd, Fth1* and *Lyz2*, particularly in cells from female APP/PS1 mice (Fig. 1c). Differentially expressed genes are shown in the volcano plot (Fig. 1d) and analysis of the mean data revealed that there was a significant genotype-related increase in genes that have been reported to characterise DAMs/ARMs like *Tyrobp, Ctsd, Ccl6* and *Trem2* ($*p < 0.05$; $**p < 0.01$; $***p < 0.001$; Fig. 1e) and these changes were validated using RT-PCR

($**p < 0.01$; $***p < 0.001$; Fig. 1f). It is notable that changes in *Tyrobp, Ctsd* and *Ccl6* were significantly greater in microglia from APP/PS1 female mice compared with males ($\S p < 0.05$; $\S\S p < 0.01$; Fig. 1e). Similar genotype- and sex-related increases were observed on other genes characteristic of DAMs/ARMs including *Cst7, Aplp2, Cd74*, and *Axl*, and genes that are upregulated in injury and/or neuroinflammatory diseases other than AD including *Gpnmb* and *Plxdc2* ($*p < 0.05$; $*p < 0.01$; $***p < 0.001$ WT v APP/PS1; $\S p < 0.05$; $\S\S p < 0.01$; $\S\S\S p < 0.001$, male vs. female APP/PS1; Supplementary Fig. 1). There were no significant changes in the 4 most-consistently described homoeostatic genes, *P2ry12, Cx3cr1, Tmem119* or *Csfr1*, although genotype-related increases were observed in *Hexb, C1qa* and *C1qc* ($*p < 0.05$; $**p < 0.01$; $***p < 0.001$) and sex-related differences in APP/PS2 mice in *Fth1* and *Lyz2* ($\S p < 0.05$; $\S\S p < 0.01$; Supplementary Fig. 1). Genotype- and sex-related changes in genes that encode proteins associated with oxidative stress, inflammation, microglial migration/motility and antigen presentation were also observed and, in many cases, changes were more marked in microglia from female APP/PS1 mice (Supplementary Fig. 2).

### Microglial morphology in APP/PS1 and WT mice is genotype- and sex-dependent.
Another marker of microglial activation, hippocampal CD68 mRNA, was increased in APP/PS1 mice compared with WT ($***p < 0.001$; Fig. 2a) and greater in female, compared with male, APP/PS1 mice ($\S p < 0.05$), while a sex-related increase, albeit not significant, was also evident in co-localisation of Iba1 and CD68 (Fig. 2b). We classified microglia according to Types 1-V (Supplementary Fig. 3) and quantified Types IV and V rod-shaped microglia, which are considered to be indicative of chronically activated cells[11] or a transitional state between ramified and amoeboid[12]. These cells were more prevalent in hippocampal sections from APP/PS1 mice ($**p < 0.01$; $***p < 0.001$; Fig. 2c) with a significantly greater proportion in tissue from female, compared with male, APP/PS1 mice ($\S\S\S p < 0.001$). Similar results were obtained in cortical tissue (Supplementary Fig. 3). We found no genotype-related differences in rod-shaped microglia in the hippocampus of 3–4-month-old mice ($8.72 \pm 1.15$ and $12.35 \pm 1.32$ in WT ($n = 6$, 3m, 3f) and APP/PS1 ($n = 5$, 3m, 2f) mice, respectively; mean ± SEM; % of total microglia).

3D surface plot reconstructions (Fig. 2d) and masks (see sample mask in Supplementary Fig. 3) were prepared to enable an analysis of morphological features and the complexity of microglia. Soma size was greater in APP/PS1 mice ($***p < 0.001$ vs. WT mice; Fig. 2e) and in male WT ($+++p < 0.001$ vs female WT mice), and also in male APP/PS1 ($\S\S\S p < 0.01$) mice compared with females. This is indicative of the more amoeboid morphology in microglia from male mice indicated in the 3D surface plot reconstructions, as is the increased circularity (Fig. 2f). Consistently, cell perimeter and area, and pixels/cell (which is indicative of ramifications) were decreased in microglia from WT male, compared with female, mice ($+p < 0.05$; $++p < 0.01$) and there was a genotype-related decrease in these measures, albeit only in females ($***p < 0.001$; Fig. 2g–i).

We observed a genotype-related decrease in numbers of branches, junctions, endpoints, triple and quadruple points and fractal dimension in APP/PS1 females compared with WT females ($**p < 0.01$; $***p < 0.001$; Supplementary Fig. 3) suggesting that genotype exerts a greater effect on cell complexity in female mice. We also report that peri-plaque Iba1[+] microglia had a reduced number of processes, decreased cell area, perimeter, diameter and pixels/cell, compared with distal, microglia

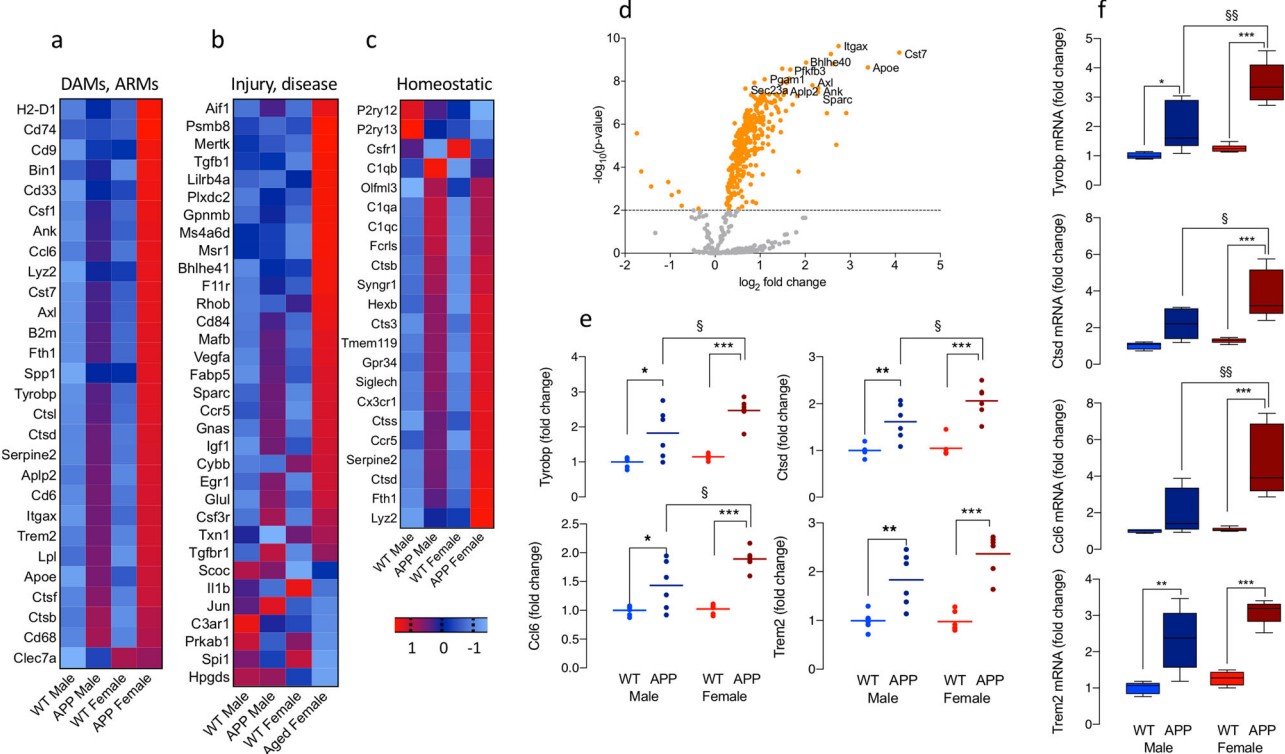

**Fig. 1 Sex-related differential expression of microglial markers in APP/PS1 and WT mice. a–c** The heat maps, generated from a Nanostring quantitative assay platform, represent the experimental groups and individual mRNA transcripts in columns and rows, respectively. Genes that have been reported to be upregulated in ARMs and/or DAMs (**a**) or in other neuroinflammatory conditions (**b**) and genes that describe the homeostatic state (**c**), are shown. Expression is displayed on a log 10 scale from blue (low expression) to red (high expression). **d** Volcano plots of mRNA expression showing significant genotype-related differences ($p < 0.01$ indicated by the dotted line) are depicted. **e, f** Significant genotype × sex interactions in *Tyrobp* and *Ccl6* ($p < 0.05$) and significant main effects of genotype ($p < 0.001$) and sex ($p < 0.05$ or $p < 0.01$) were identified in mean data from Nanostring analysis and RT-PCR for *Tyrobp, Ctsd, Ccl6*, and *Trem2*. Post hoc analysis revealed significant genotype-related increases (*$p < 0.05$; **$p < 0.01$) and significant increases in *Tyrobp*, *Ctsd*, and *Ccl6* in microglia from female, compared with male, APP/PS1 mice (§$p < 0.05$; §§$p < 0.01$; §§§$p < 0.001$) and in several other indicators of microglial activation as indicated in Supplementary Fig. 1. Data, expressed as means ± SEM ($n = 5$ (PCR data) or 6 (Nanostring data)), were analysed by 2-way ANOVA and Tukey's post hoc multiple comparison test. The changes in **e** and **f** are relative to values in WT males. Additional related data are presented in Supplementary Figs. 1 and 2.

($p < 0.001$; Fig. 3a, b) and, in distal microglia, these measures were decreased in male, compared with female, mice (§§§$p < 0.001$) indicating that cell complexity was reduced in males.

**Microglia from female APP/PS1 mice shift their metabolism towards glycolysis.** When microglia are activated in APP/PS1 mice, they shift their metabolism towards glycolysis[8]. Here, we used multiplexed gene expression analysis (NanoString nCounter) to investigate differentially expressed genes related to metabolism in microglia. The heat map (Fig. 4a) identifies striking genotype-related changes in genes related to glucose metabolism in isolated microglia, that are particularly marked in female APP/PS1 mice. Significant upregulation in glycolytic enzymes, *Hk2, Pfkfb3, Gapdh, Pgk1* and *Pgam1*, was evident in cells from APP/PS1, compared with wild-type mice (*$p < 0.05$; **$p < 0.01$; ***$p < 0.001$; Fig. 4b) and the changes in *Gapdh, Pgk1* and *Pgam1* were significantly greater in female APP/PS1 mice compared with males (§$p < 0.05$; §§$p < 0.01$).

Analysis of metabolic profile by Seahorse showed a significant genotype-related increase in ECAR in female mice (***$p < 0.001$; Fig. 4c) and glycolysis was significantly increased in microglia from female, compared with male, APP/PS1 mice (§§$p < 0.01$; Fig. 4d). This was mirrored by changes in lactate as revealed by

metabolomic analysis (§$p < 0.05$; Fig. 4e) although no genotype- or sex-related changes were observed in intermediate metabolites of glycolysis (Fig. 4f). There was no significant change in the tricarboxylic acid (TCA) intermediates citrate, α-ketoglutarate or *cis*-aconitate, but a genotype-related decrease in fumarate and malate was detected in cells from male mice (*$p < 0.05$; Supplementary Fig. 4) and succinate, which is associated with inflammatory cytokine production in macrophages, was increased in microglia from female, compared with male, APP/PS1 mice (§$p < 0.05$). Another clear sex-related difference in metabolism was the significant increase in several amino acids in microglia from female, compared with male, APP/PS1 mice (Supplementary Fig. 5) that are derived from intermediates of glycolysis and the TCA.

An increase in expression of PFKFB3 consistently accompanies increased glycolysis in microglia[8,9] and here, a main effect of genotype was observed in PFKFB3 staining in Iba1+ cells (*$p < 0.05$; Fig. 5a, b), although post hoc analysis revealed no significant changes. However, PFKFB3 is activated only when it translocates from the nucleus to the cytosol and cytosolic PFKFB3 staining was significantly increased in cells from female APP/PS1 mice compared with males (§§$p < 0.01$; Fig. 5c, d) and this was accompanied by a significant decrease in nuclear staining (§§$p < 0.01$) and a significant reduction in nuclear:cytosolic PFKFB3 (§$p < 0.05$) indicating

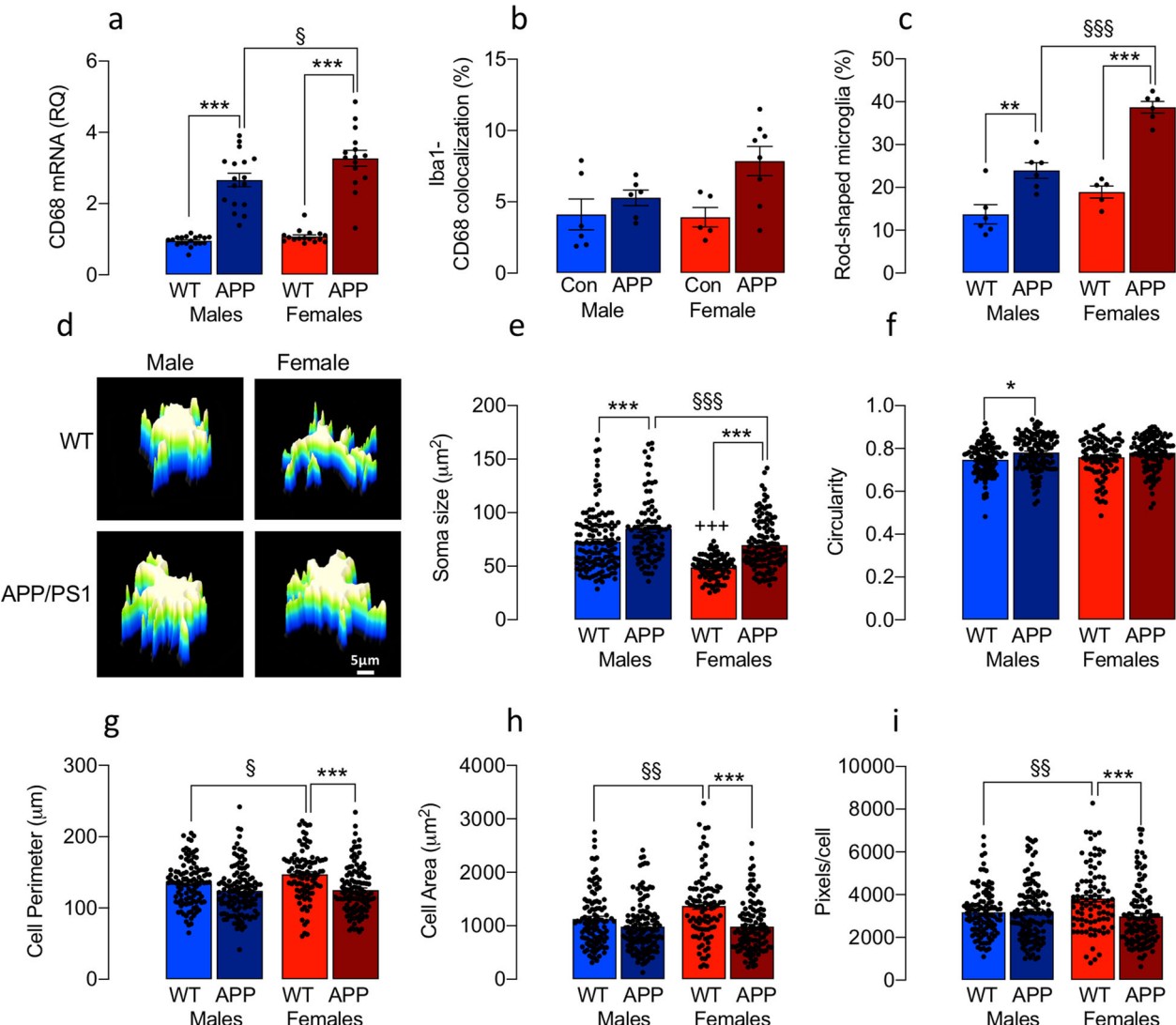

**Fig. 2 Evidence of sex-related changes in microglial morphology in APP/PS1 and WT mice. a–c** CD68 mRNA (**a**), the co-localisation of Iba1⁺ CD68⁺ pixels (**b**) and the proportion of rod-shaped microglia (**c**) were increased in the hippocampus (and cortex, see Supplementary Fig. 2) of APP/PS1, compared with WT, mice (***$p < 0.001$) and a further increase was observed in female, compared with male, APP/PS1 mice (§$p < 0.05$; §§§$p < 0.001$). **d** 3D surface plot reconstructions show that male cells adopt an amoeboid morphology (scale bar = 5 µm). **e** A genotype-related increase in soma size was evident in male and female mice (***$p < 0.001$; **d**) but soma size was reduced in female WT and APP/PS1 mice compared with male counterparts (+++$p < 0.001$; §§§$p < 0.001$, respectively). **f** A significant main effect of genotype in circularity ($p < 0.001$) was observed and the mean value was significantly increased in male APP/PS1, compared with WT, mice (*$p < 0.05$). **g–i** Cell perimeter, area and pixels/cell were increased WT male, compared with the female mice (+$p < 0.05$; ++$p < 0.01$). Genotype-related decreases were observed in female mice (***$p < 0.001$) and changes in cell complexity were identified mainly in microglia from female mice (Supplementary Fig. 2). Data, expressed as means ± SEM ($n = 5$ or 6 mice/group with analysis of between 96 and 128 cells), were analysed by 2-way ANOVA and Tukey's post hoc multiple comparison test. In the case of CD68 mRNA, data were retrospectively calculated from several previous experiments and assessed by sex ($n = 15$) (APP female; 16 WT female; 17 APP male; 19 WT male).

preferential enzyme activation, consistent with the glycolytic signature, in microglia from female APP/PS1 mice.

**Differential effect of sex on microglial function.** Glycolytic microglia are functionally impaired with reduced phagocytic and chemotactic capability[8,9]. Phagocytosis was assessed by evaluating the uptake of Aβ into isolated microglia from male and female mice; the percentage of Aβ⁺ Iba⁺ cells was reduced to a greater extent in microglia from female APP/PS mice compared with WT (*$p < 0.05$; Fig. 6a). Interestingly the relatively greater functionality in microglia from male mice is also suggested by the characteristic amoeboid morphology and supported by increased

CD68 immunoreactivity in Iba1⁺ cells, increased phagocytosis of Aβ and fluorescently labelled beads in amoeboid, compared with non-amoeboid microglia, and also increased CD68 staining in amoeboid microglia (Supplementary Fig. 6). There was a sex-related increase in ThioS-stained Aβ plaque number and area in the hippocampus (*$p < 0.05$; Fig. 6b), which may, in part, be explained by the decreased phagocytosis. We found that phago-lysosomal loading with Aβ was significantly greater in females (*$p < 0.05$; Fig. 6c) suggesting that lysosomal processing of Aβ may be decreased. Consistent with this is the data from the NanoString analysis, which indicated that lysosomal genes appeared to be upregulated in microglia prepared from female APP/PS1 mice compared with the other groups (Fig. 6d)

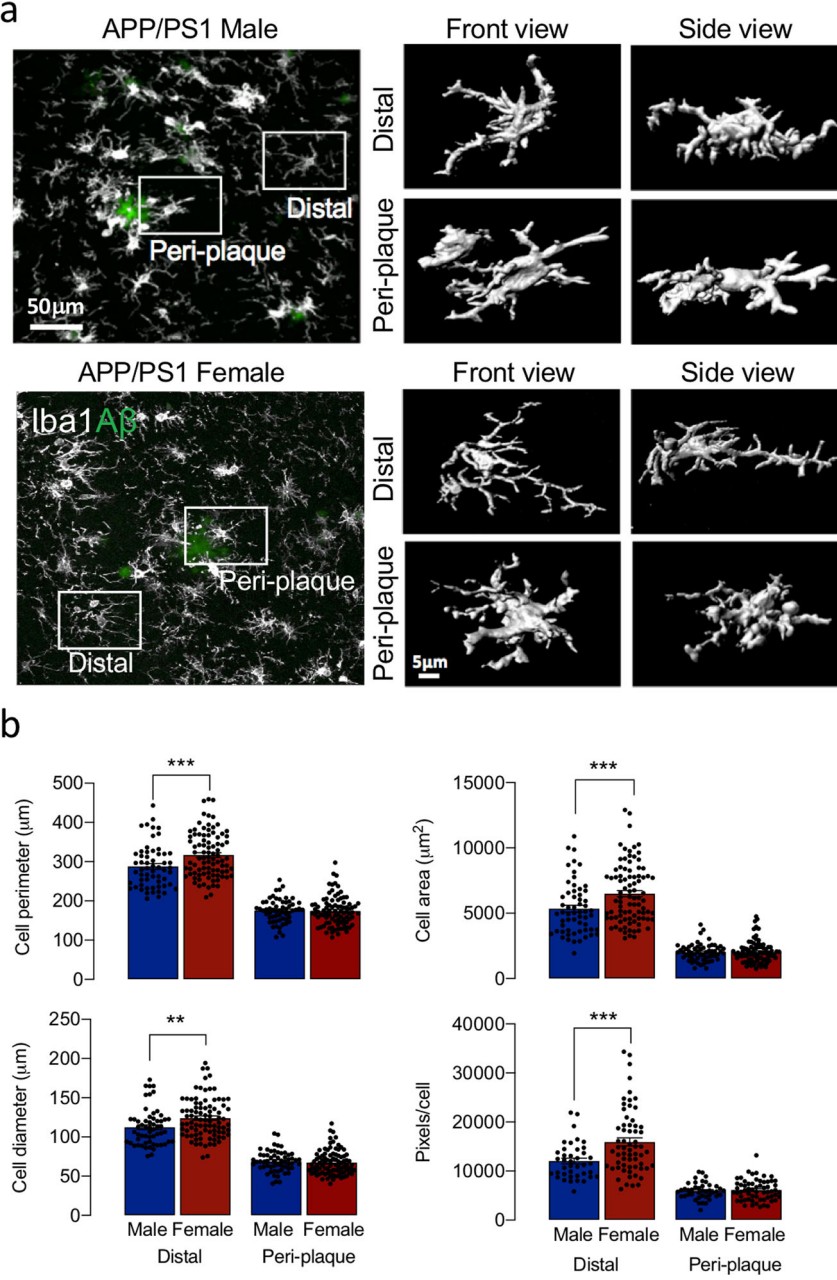

**Fig. 3 Evidence of sex-related differences in morphology of microglia distal from plaques. a** 3D reconstructions showed differences in morphology in peri-plaque and distal microglia (scale bars = 50 and 5 μm in the main image and high-magnification image, respectively). **b** Cell perimeter, area, diameter (59–89 cells analysed) and pixels/cell (42–63 cells analysed) were markedly reduced in peri-plaque compared with distal microglia and these measures were increased in distal microglia from female, compared with male, mice ($^{§§§}p < 0.001$). Data, expressed as means ± SEM ($n = 5$ or 6 mice/group), were analysed by 2-way ANOVA and Tukey's post hoc multiple comparison test.

including genes that encode lysosomal proteins like *Lamp2*, genes encoding for proteins that play a role in lysosomal function like *Myo5A* and *Rab3A* and genes encoding for degrading enzymes like *Ctsl*, *Cla*, *Galc*, *Ppt1* and *Srgn* ($^{§}p < 0.05$; $^{§§}p < 0.01$; Fig. 6e)

**Microglial morphology and amyloidosis in AD post-mortem tissue are sex-related.** We compared microglia in post-mortem cortical tissue from male and female individuals with AD and age-matched controls. Microglia from male AD brain showed evidence of process retraction (Fig. 7a) with an amoeboid appearance (Fig. 7b), whereas cells from female brains were more complex and variable in morphology (Fig. 7a) with many rod-

shaped microglia (Fig. 7b). The representative masks and 3D representations (Fig. 7c), which illustrate these morphological differences, were used to analyse morphological features and the data indicate significant disease x sex interactions in Feret's diameter, perimeter and density ($p < 0.05$; Fig. 7d). Post hoc analysis revealed a significant increase in density and decreases in perimeter and Feret's diameter in microglia from male AD patients compared with male controls, with generally smaller changes in females ($*p < 0.05$; $**p < 0.01$; $***p < 0.001$). Significant differences between male, compared with the female, AD patients were also observed in these measures ($^{§§§}p < 0.001$). Although changes in circularity broadly paralleled the changes in density, perimeter and Feret's diameter, these did not reach statistical significance.

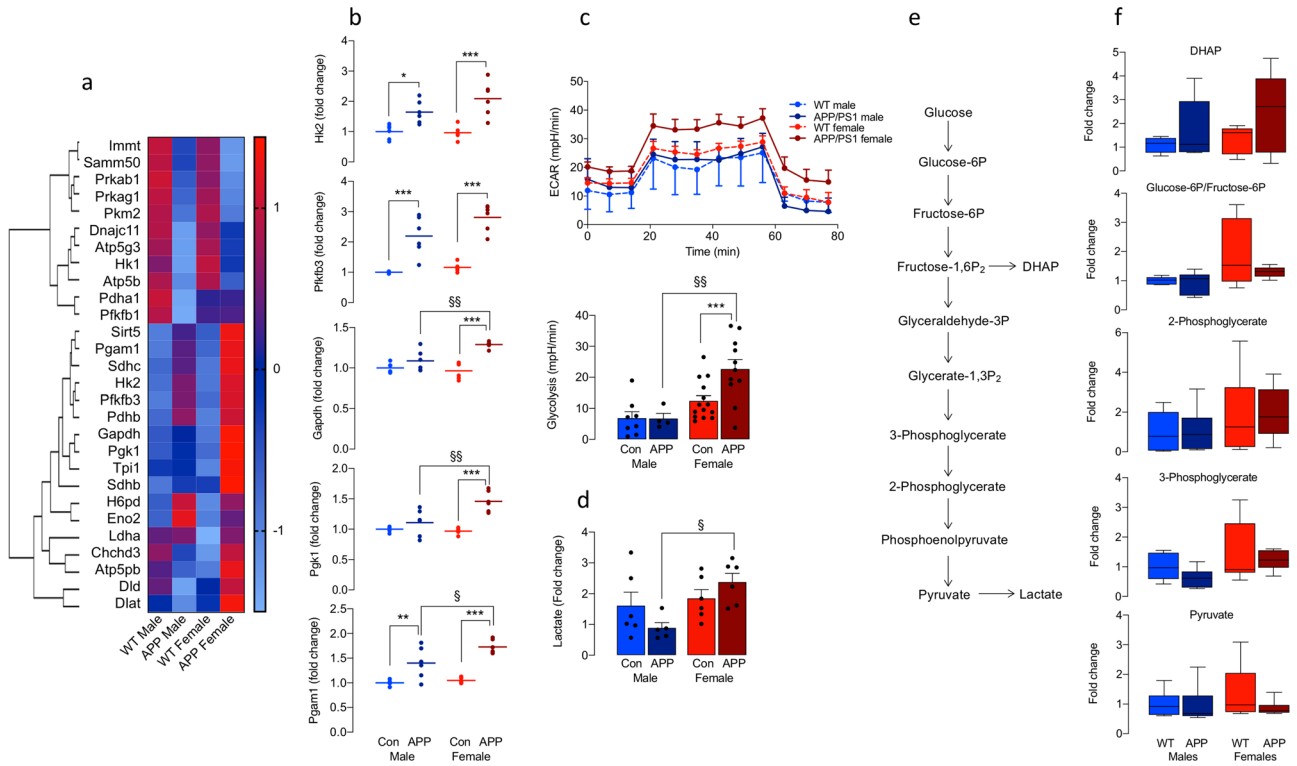

**Fig. 4 Microglia from female APP/PS1 mice shift their metabolism towards glycolysis. a, b** NanoString analysis indicated that *Hk2, Pfkfb3, Gapdh, Pgk1* and *Pgam1* were upregulated in cells from APP/PS1, compared with WT, mice (*$p < 0.05$; **$p < 0.01$; ***$p < 0.001$; $n = 6$) with some changes significantly greater in female APP/PS1 mice compared with males (§$p < 0.05$; §§$p < 0.01$). Expression is displayed on a log 10 scale from blue (low expression) to red (high expression). **c** ECAR was increased in microglia from female APP/PS1 mice with a significant increase in glycolysis in microglia from female APP/PS1 ($n = 11$), compared with female WT ($n = 14$), mice (***$p < 0.001$) and male APP/PS1 mice (§§$p < 0.01$; $n = 4$) and also male WT mice ($n = 8$). **d** Lactate was increased in microglia from female APP/PS1 mice ($n = 6$) compared with male APP/PS1 mice (§$p < 0.05$; $n = 5$). **e, f** No genotype- or sex-related changes were observed in other intermediate metabolites of glycolysis ($n = 6$ except for DHAP where $n = 5$). Minimal changes were observed in intermediate metabolites of the TCA (Supplementary Fig. 3) although marked sex-related differences were identified in amino acids generated from intermediate metabolites of glycolysis and the TCA (Supplementary Fig. 3). The changes in **b, d** and **f** are relative to values in WT males. Data, expressed as means ± SEM, were analysed by 2-way ANOVA and Tukey's post hoc multiple comparison test.

The plaque area, assessed by Congo red staining, was significantly increased in tissue sections from female AD patients compared with males (*$p < 0.05$; Fig. 8a). In contrast, amyloid staining in the vasculature was significantly greater in sections prepared from male AD patients compared with females (***$p < 0.001$; Fig. 8b). Because of our findings in mice, we used CD68 as a proxy marker of phagocytic microglia, and asked whether CD68 immunoreactivity was more pronounced in the parietal cortex from male, compared with the female, AD patients. The data revealed an increase in CD68 staining in the parietal cortex of male, compared with female, AD patients (*$p < 0.05$; 1-tailed *t*-test; Fig. 8c). The pattern of staining was different in male and female AD patients with apparently greater clustering of CD68 in sections from female patients.

## Discussion

There is virtual unanimity that microglia are inappropriately activated in AD and models of AD but analysis of sex-related differences has not been systematically assessed. The focus of this study was to address sex-related differences and the significant findings are that there is a preferential upregulation in genes linked with microglial activation in cells from female, compared with male, APP/PS1 mice accompanied by evidence of sex-related differences in microglial morphology, metabolism and function that we propose to contribute to the amyloid pathology. The sex-related changes in microglial morphology and amyloidosis

observed in the animal model were reflected in post-mortem tissue from AD patients.

We report a genotype-related upregulation of several genes indicative of microglial activation and, importantly, this upregulation was markedly greater in microglia from female APP/PS1 mice compared with males. These genes include *Tyrobp, Ctsd, Axl, Cst7* and others, described as DAMs and/or ARMs, that are increased in 5XFAD and App[NL-G-F] models of AD[5,6]. The novel finding is that there is a differential enhancement in the expression of several genes that are indicative of an activated or diseased state in microglia from female mice. A number of the genes that exhibited sex-related differences were linked with altered risk of AD, for example, *Apoe, Bin1, Trem2* and *Cd33*[10], as well as several members of the cathepsin family of cysteine proteinases. In addition, genes that are upregulated following injury and/or in models of diseases amyotrophic lateral sclerosis and multiple sclerosis[1,3]. Several genes indicative of oxidative stress and inflammation were also increased in female APP/PS1 mice compared with male APP/PS1 mice, including *Ubqln1, App* and *Hmox* that play a role in regulating oxidative stress-induced signalling, *Nkiras1*, which modulates NFκB signalling, and *Mef2a* and *Panx1*, which regulate microglial surveillance and migration, respectively. Marked sex-related differences were also observed in genes like *Itgam, Aif1* and *Vim*, which impact microglial motility and *Atg3, Ctsl* and *Fcgr1*, which are involved in antigen presentation. The ramifications of such marked changes in gene expression may suggest that microglia from females are more

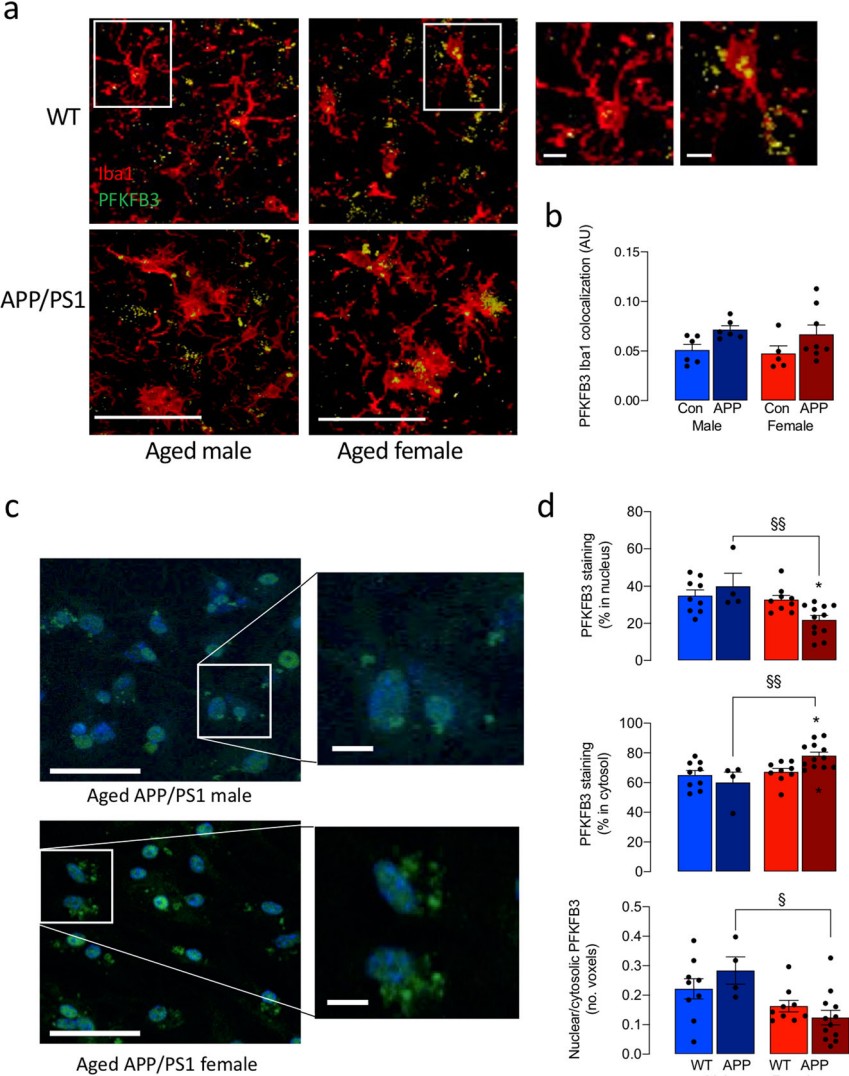

**Fig. 5 Cytosolic PFKFB3 is increased in microglia from female APP/PS1 mice. a** Representative images of PFKFB3 staining in Iba1+ microglia in sections from the hippocampus. (Scale bar = 100 μm; magnified image, 20 μm). **b** A genotype-related increase in PFKFB3 staining was found in sections prepared from APP/PS1, compared with WT, mice ($p < 0.05$; main effect of genotype) though post hoc analysis identified no significant changes. Values are presented as means ± SEM ($n = 6$ for male WT and APP/PS1 mice, $n = 5$ and 7 for female WT and APP/PS1 mice, respectively). **c** Representative images of PFKFB3 staining in isolated microglia. Scale bars: main figure, 50 μm; magnified image, 10 μm. **d** The percentage of PFKFB3 staining in the cytosol was significantly increased while nuclear staining was significantly decreased in microglia from female APP/PS1 mice compared with male APP/PS1 mice, and the ratio nuclear:cytosolic staining (expressed as voxels) was similarly decreased (§$p < 0.05$; §§$p < 0.01$; $n = 9, 4, 9, 12$ for male WT and APP/PS1 and male WT and APP/PS1 mice, respectively). Data, expressed as means ± SEM, were analysed by 2-way ANOVA and Tukey's post hoc multiple comparison test.

primed for change than microglia from males and potentially more dysfunctional and less neuroprotective, perhaps explaining the sex-related difference in behaviour[13].

To this point, analysis of sex-related differences in microglia in AD models has been largely unexplored although one study suggested that female mice display a somewhat faster progression towards the profile that identifies ARMs than males[6]. Analysis of age-related changes in genes reflecting microglial activation indicated an upregulation that was more marked in hippocampal tissue[2] and in RiboTag-derived microglial transcripts[14] of females compared with males.

Morphological features of microglia have often been used to describe activation state. Rod-shaped microglia are characteristic of age[12,15] as well as chronic disorders of the CNS including AD and subacute sclerosing panencephalitis, where they are considered to be a reflection of a persistently activated cell[11]. Thus one interpretation of the data is that the increased number of

these cells in female APP/PS1 mice reflects a persistently activated state that is not evident in males. However, others have suggested that the cells represent a transitional state between ramified and amoeboid[12]; this interpretation may be consistent with the increased plaque deposition in females because the transformation to amoeboid is dependent on developing pathology. It is unclear whether the appearance of rod-shaped microglia precede amyloidosis although no genotype-related change was observed in hippocampal tissue prepared from 3-month-old APP/PS1 mice compared with WT mice.

Microglia from male WT mice were more amoeboid than female WT mice, and consistent with this, soma size was greater and perimeter and cell area were decreased identifying sex-related differences in morphology in aged mice in the absence of amyloid pathology. The importance of this is that amoeboid cells are considered to be phagocytic and mobile[16–18], which is supported by the evidence that amoeboid cells take up more Aβ and

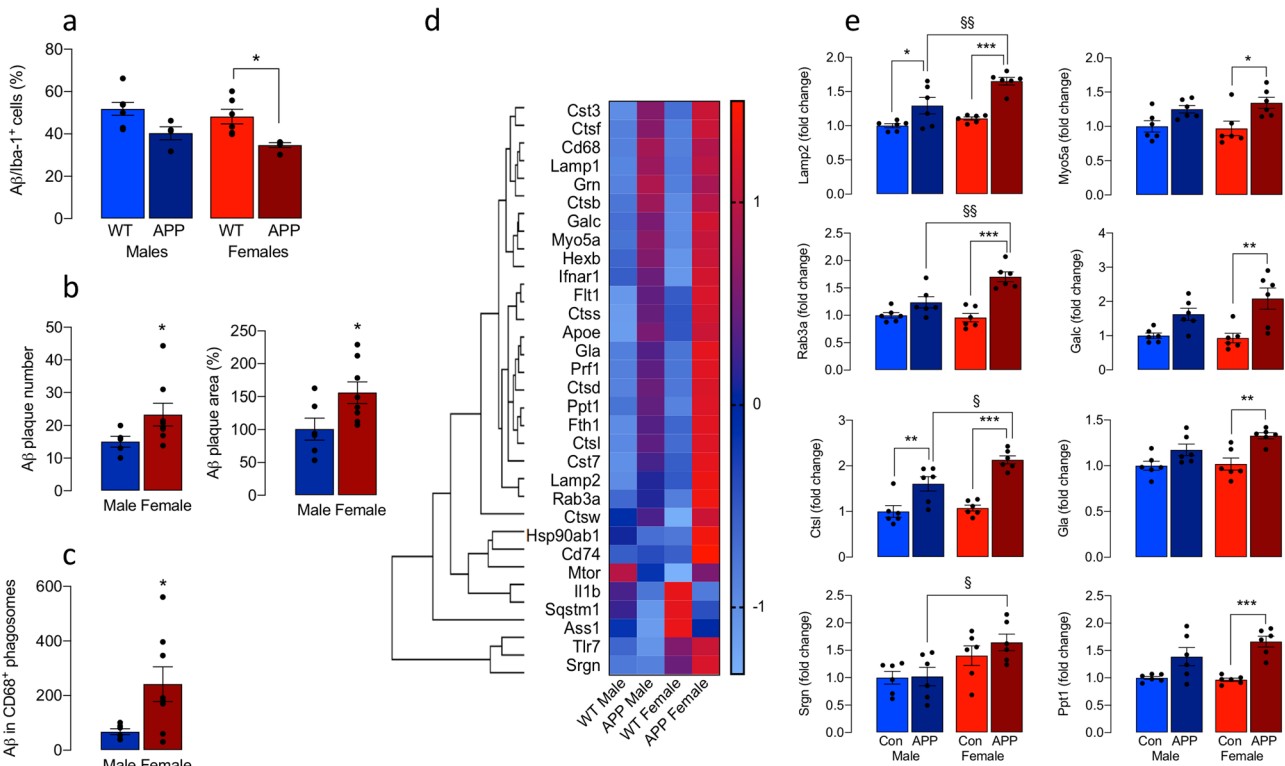

**Fig. 6 Differential effect of sex on microglial function. a** Aβ uptake into isolated microglia from female APP/PS1 mice was significantly reduced compared with microglia from WT mice (*p < 0.05; n = 7, 4, 6, 5 for male WT and APP/PS1 and male WT and APP/PS1 mice, respectively). **b** ThioS-stained Aβ plaque number and area were significantly increased in hippocampal sections from female, compared with male, APP/PS1 mice (*p < 0.05; n = 5 and 8 for male and female mice, respectively). **c** Phagolysosomal loading with Aβ was significantly greater in microglia from female APP/PS1 mice compared with males (*p < 0.05; n = 6 and 8 for male and female mice, respectively). **d, e** NanoString analysis indicated that lysosomal genes were upregulated in microglia prepared from female APP/PS1 mice compared with the other groups (**d**). Analysis of the mean data, relative to values in WT males, indicated that there were significant increases in *Lamp2, Myo5A, Rab3A, Ctsl, Cla, Galc, Ppt1* and *Srgn* in microglia from female APP/PS1 mice compared with female WT mice (*p < 0.05; **p < 0.01; ***p < 0.001) and compared with male APP/PS1 mice (§p < 0.05; §§p < 0.01; n = 6; **e**). Data, expressed as means ± SEM, were analysed by 2-way ANOVA and Tukey's post hoc multiple comparison test except for **b** and **c** when the Student's t-test for independent means was used to evaluate data.

fluorescently labelled beads compared with non-amoeboid cells. This is consistent with the proposal that upregulation of genes indicative of an activated/disease state exhibit reduced function[8,9]. Analysis of morphology in microglia that were distal from plaques indicated a more amoeboid morphology in males perhaps indicating the increased phagocytic capacity in these cells and reduced pathology.

We demonstrate that phagocytosis of Aβ by isolated microglia is reduced in female, compared with male, APP/PS1 mice. Interestingly, it has been proposed that phagocytosis is suppressed when *Cst7* is upregulated[14] and this is significant because here we show that *Cst7* is markedly upregulated in microglia from female, compared with male, APP/PS1 mice. The sex-related decrease in phagocytosis is associated with an increase in amyloid load in sections prepared from female APP/PS1 mice and this supports previous findings of increased amyloidosis in female APP/PS1 mice[13,19]. However, despite the change in phagocytosis, Aβ loading of CD68[+] phagolysosomes in Iba1[+] cells from female APP/PS1 mice was increased compared with males, and several genes that reflect aspects of lysosomal function including degradative enzymes were also increased in a sex-related manner. It is known that lysosomal acidification is defective in APP/PS1 mice[20] and this results in compromised amyloid degradation, contributing to amyloid pathology, although sex-related changes in lysosomal function have not been assessed to date.

We have proposed that the compromise in phagocytic function in microglia from APP/PS1 mice is a result of altered metabolism, which is closely correlated with inflammation[8]. A consistent finding from our lab is that microglia which are inflammatory switch their metabolism towards a more glycolytic state[8,9,21,22] and when this less efficient metabolic profile persists, the phagocytic function becomes compromised. Therefore we predicted that microglia from female APP/PS1 mice would switch their metabolism towards glycolysis and the evidence indicates that this is the case. There was a marked sex-related difference in the metabolic profile, with increased ECAR, increased mean glycolysis and lactate in microglia from female APP/PS1 mice. This is an important advance, indicating that the previously described genotype-related changes in microglial activation and metabolism are attributable more to females than males, in which there is a minimal shift towards glycolysis. Thus ATP production is not compromised in microglia from male APP/PS1 mice, which may provide an explanation for the maintenance of phagocytic function.

PFKFB3 is a key regulator of glycolysis[23,24] and an increase in expression of PFKFB3 in microglia is observed in glycolytic cells[8,9,22]. Here the genotype-related upregulation of several glycolytic enzymes including PFKFB3, together with the translocation of PFKFB3 from the nucleus to the cytosol, that was clearly upregulated in microglia from female APP/PS1 mice, indicates

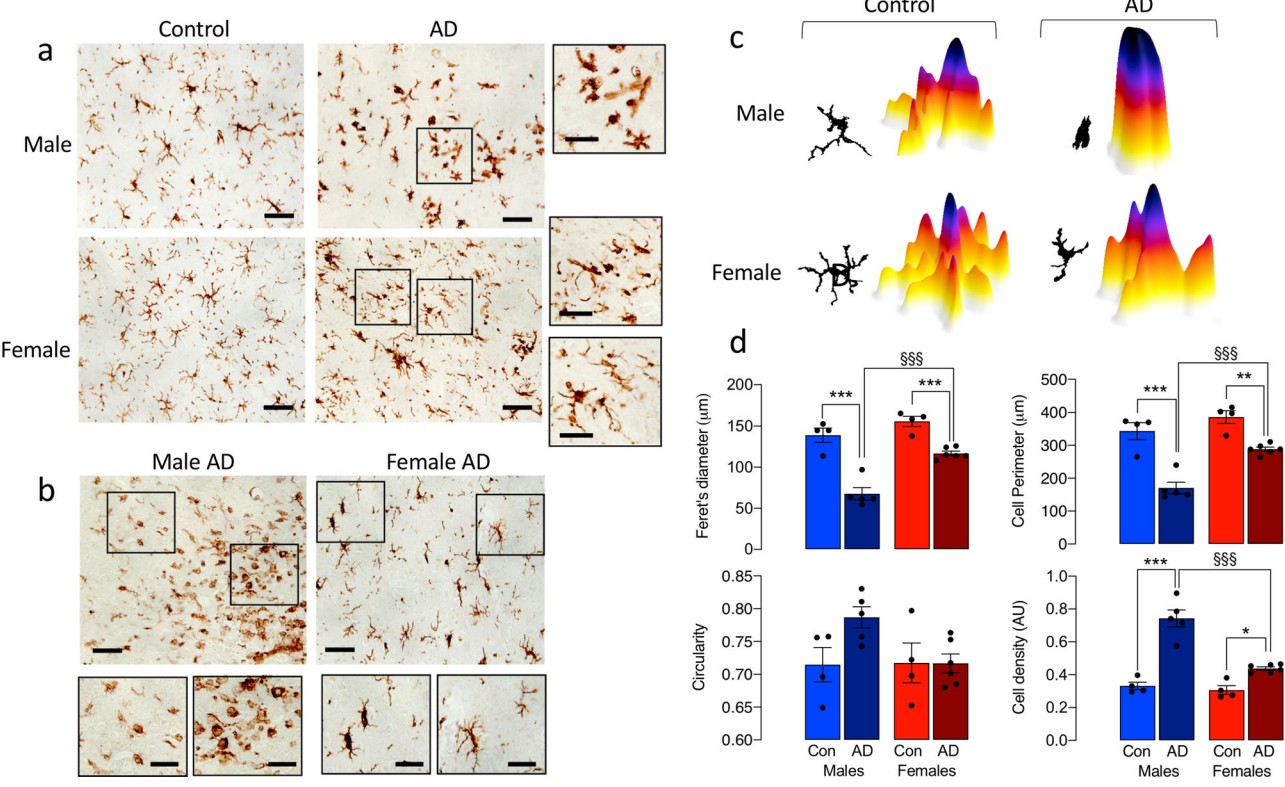

**Fig. 7 Changes in microglial morphology in post-mortem parietal cortical tissue from AD patients are sex-related. a, b** Representative images of DAB-stained microglia, showing marked process retraction (**a**; scale bar = 100 μm; magnified image, 50 μm) and a preponderance of amoeboid microglia (**b**; scale bar = 200 μm; magnified image, 50 μm) in sections of parietal cortex from male AD patients compared with females, in which many rod-shaped microglia were identified. **c** Representative masks and 3D representations were used to analyse morphological features. **d** Significant disease × sex interactions in circularity, perimeter, Feret's diameter and cell density were observed ($p < 0.05$). Post hoc analysis revealed significant increases in circularity and cell density and significant decreases in cell perimeter and Feret's diameter in microglia from male AD patients compared with controls (**$p < 0.01$; ***$p < 0.001$) and also in male, compared with female, AD patients ($§p < 0.05$). Data, expressed as means ± SEM ($n = 4$ for controls or 5 for AD), were analysed by 2-way ANOVA and Tukey's post hoc multiple comparison test.

that activation of PFKFB3 may drive the sex-related difference in glycolysis. The human PFKFB3 gene is located on chromosome 10, on which AD-susceptibility loci have been identified[25] although explicit data indicating PFKFB3 variants in the disease have not been reported. The same group reported an association between AD and *GAPD* genes[26], which is interesting since the preferential sex-related increase in glycolytic enzymes in microglia from females in the metabolomic analysis was accompanied by a similar increase in GAPDH.

There were few genotype- or sex-related changes in TCA cycle metabolites but succinate was increased in cells from females, which is significant because succinate is an inflammatory molecule[27,28] contributing to the production of inflammatory cytokines including IL-1β[29], which is increased in the brain of APP/PS1 mice[30].

A significant finding in this study is that the sex-related differences extended to human post-mortem tissue. Specifically, analysis of plaque coverage showed an increase in sections of the cortex from female AD patients compared with male AD patients, contrasting with the increased amyloid staining in vasculature in males. Furthermore, microglia from male AD patients were amoeboid with little evidence of heterogeneity in morphology and this contrasted sharply with brains from female AD patients where there was marked heterogeneity with few amoeboid cells, some ramified cells, and numerous rod-shaped microglia, which were also observed in greater numbers in sections from female APP/PS1 mice. However, in contrast to the findings in WT mice,

there were few amoeboid microglia in tissue from age-matched control males.

The parallel increases in amoeboid microglia and increased CD68 immunoreactivity in microglia from male AD patients is consistent with previous data that suggest amoeboid cells are phagocytic[18]. One possible interpretation of the findings is that there may be a greater preservation of function, particularly responsiveness to pathological stimuli, in microglia from male, compared with the female, AD patients. In this context, a recent meta-analysis revealed that, when all variables were accounted for, the cognitive abilities of women with AD patients were worse than men with AD[31].

We have identified marked sex-related differences in microglial activation, morphology, metabolism and function in APP/PS1 mice and we propose that these linked changes contribute to the more marked amyloid pathology, reduced neuroprotection and more profound cognitive impairment in females.

## Methods

**Animals**. Male and female transgenic mice that overexpress mutant amyloid precursor protein (APP$_{swe}$) and presenilin 1 (PSEN1$_{ΔE9}$; APP/PS1 mice, B6;C3-Tg (APP$_{swe}$PSEN1$_{ΔE9}$)85Dbo/Mmjax MMRRC (RRID:MMRRC_034829-JAX)) and littermate controls were used in these studies. Animals were housed under controlled conditions (20–22 °C, food and water ad lib) and maintained under veterinary supervision and were 16–18 months old at the time of the experiments. The animal studies were conducted under license from the Health Products Regulatory Authority (HPRA), Ireland (Project licence numbers AE19136/P035 and

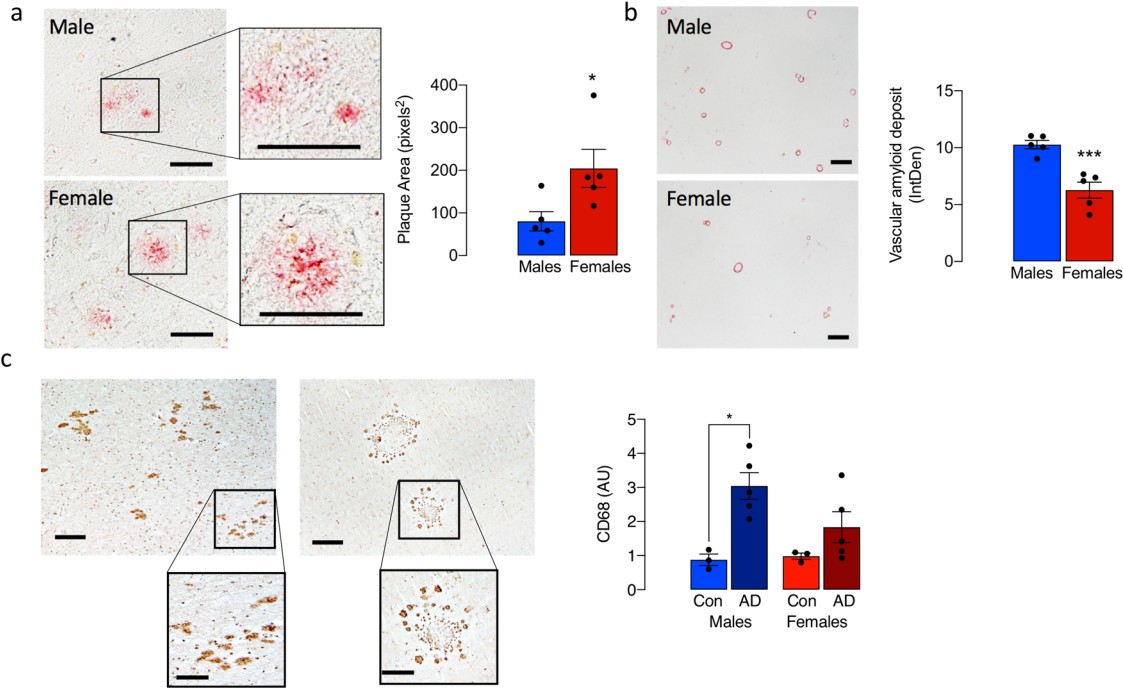

**Fig. 8 Amyloid accumulation is significantly greater in post-mortem tissue from female compared with male AD patients. a** Congo red staining in the parenchyma was more pronounced in parietal cortical tissue from female, compared with male, AD patients and plaque area was significantly increased (*$p < 0.05$; $n = 5$; scale bar = 100 μm). **b** Vascular amyloid staining was significantly greater in sections prepared from male AD patients compared with females (***$p < 0.001$; $n = 5$; scale bar = 400 μm). **c** CD68 immunoreactivity was significantly greater in parietal cortex from male, compared with the female, AD patients (*$p < 0.05$, 1-tailed $t$-test; $n = 4$ except for AD females where $n = 5$; scale bar = 100 μm; magnified image, 50 μm). Data, expressed as means ± SEM ($n = 5$), were analysed by the Student's $t$-test (1-tailed $t$-test for CD68 and 2-tailed for amyloid staining).

**Table 1 Human samples.**

| Donor | Sex | Age | Brain weight | Pmd | Braak Score | Brain area |
|---|---|---|---|---|---|---|
| *Non-demented controls* | | | | | | |
| 1995-006 | M | 81 | 1274 | 505 | 1 | H & PC |
| **1996-052** | **M** | **73** | **1500** | **550** | **2** | **H** |
| 1998-126 | M | 71 | 1345 | 360 | 2 | H & PC |
| 2017-016 | M | 72 | 1385 | 260 | 2 | H & PC |
| 1995-110 | F | 81 | 1022 | 1335 | 1 | H & PC |
| 2011-028 | F | 81 | 1075 | 265 | 1 | H & PC |
| 2015-034 | F | 82 | 1318 | 465 | 1 | H & PC |
| 2015-087 | F | 75 | 1305 | 550 | 1 | H & PC |
| **1997-163** | **M** | **74** | **1328** | **353** | **2** | **PC** |
| *Alzheimer's disease* | | | | | | |
| 1994-086 | M | 75 | 1098 | 330 | 5 | H & PC |
| 2000-042 | M | 78 | 1208 | 465 | 5 | H & PC |
| 2001-092 | M | 79 | 1118 | 305 | 5 | H & PC |
| 2007-025 | M | 82 | 1182 | 315 | 5 | H & PC |
| 2007-078 | M | 77 | 1104 | 270 | 5 | H & PC |
| 1992-025 | F | 80 | 1046 | 265 | 5 | H & PC |
| 1992-069 | F | 78 | 889 | 265 | 5 | H & PC |
| 1992-100 | F | 77 | 1016 | 252 | 5 | H & PC |
| 1994-083 | F | 75 | 960 | 255 | 5 | H & PC |

Details of hippocampal and parietal samples obtained from the Netherlands brain bank. Samples were obtained from the same individuals except Donor 1996-052 (hippocampus) was replaced by Donor 1997-163 (parietal cortex); these samples are highlighted in bold text.
*Pmd* post-mortem delay, *H* hippocampus, *PC* parietal cortex

AE19136/P113) in accordance with EU regulations and with local ethical approval (Animal Research Ethics Committee, Trinity College Dublin).

**Human samples**. Paraffin-embedded human brain tissue (Table 1) was obtained from the Netherlands Brain Bank (NBB). Consent from donors was obtained by the NBB and this meets all the current legal and ethical guidelines for brain autopsy, tissue storage and use of tissue for scientific research worldwide. Experiments were undertaken with local ethical approval (Faculty of Health Sciences Research Ethics Committee, Trinity College Dublin).

**Preparation of mouse tissue**. Mice were anaesthetised with sodium pentobarbital (Euthanimal), transcardially perfused with saline and the brain was dissected free

**Table 2 nCounter metabolic custom panel.**

| Customer identifier | Accession |
|---|---|
| Gapdh | NM_001001303.1 |
| H6pd | NM_173371.3 |
| Hk2 | NM_013820.3 |
| Ldha | NM_010699.2 |
| Pfkfb1 | NM_008824.2 |
| Pfkfb3 | NM_133232.2 |
| Pkm2 | NM_011099.2 |
| Sdhb | NM_023374.3 |
| Sdhc | NM_025321.3 |

and, in different studies, used to isolate microglia for analysis by NanoString, immunostaining, ex-vivo metabolic analysis, metabolomics and assessment of phagocytic capacity, or prepared for immunohistochemical (IHC) or gene expression analyses. For IHC, brains were perfused with cold PBS and PFA (4%), incubated in PFA (4%, 24 h) and stored in sucrose (30%) for preparation of coronal sections (40 μm). Sections were stored (30% ethylene glycol, 30% sucrose in PBS, −20 °C) and later used for staining.

**Preparation of microglia.** Isolated microglia were prepared from mouse brains as previously described[32]. Briefly, brain tissue was homogenised (Gentle-MACS Dissociator and the Adult Brain Dissociation Kit, Miltenyi Biotec, UK), filtered and washed with Dulbecco's phosphate-buffered saline (D-PBS; PBS containing calcium (100 mg/l), magnesium (100 mg/l), glucose (1000 mg/l), and pyruvate (36 mg/l)) and samples were centrifuged (3000×g, 10 min) to remove cell debris. Supernatant samples were centrifuged (300×g, 10 min), the pellet was resuspended in D-PBS, and the resultant microglia were incubated with CD11b microbeads (Miltenyi Biotec. UK) and magnetically separated using the QuadroMACS separator (Miltenyi Biotec, UK) according to the manufacturer's instructions. Samples were resuspended in PBS containing 0.5% foetal bovine serum and centrifuged (300×g, 10 min) and the pellet was resuspended in culture media: Dulbecco's modified Eagle's medium (DMEM-F12). Cells were seeded in 48-well plates (60,000 cells/well; final volume 200 μl) for immunostaining, in 24-well plates (80,000 cells/well; final volume 200 μl) for analysis of phagocytosis, in 96-well plates (60,000 cells/well) for metabolic profile assessment using the SeaHorse Extracellular Flux (XF96) Analyser (SeaHorse Bioscience, US), in 12-well plates (200,000 cells/well and 400,000 cells/ well) for metabolomics and NanoString analysis, respectively. In all cases, cells were cultured for 5 days and media supplemented with macrophage colony-stimulating factor (M-CSF; 100 ng/ml; R&D Systems, UK) and granulocyte-macrophage colony-stimulating factor (GM-CSF; 100 ng/ml; R&D Systems, UK) was changed on alternate days.

**Gene expression analysis.** RNA was isolated from microglia using NucleoSpin RNAII kit (Macherey-Nagel, Duren, Germany) and samples were assessed in nCounter hybridisation reactions containing NanoString Reporter CodeSet, capture ProbeSet and hybridisation buffer added with Proteinase K according to the manufacturer's instructions (nCounter Mouse Glial Profiling Panel plus a custom panel (Table 2)). Briefly, samples that were stored at −80 °C were thawed and unamplified RNA (30 ng) was used per hybridisation assay (overnight, 65 °C in thermal cycler). After hybridisation, reactions were loaded on a nCounter MAX Analysis system. Data analysis was performed using nSolver software; negative and positive controls facilitated technical normalisation, adjusting for variability arising from differences in hybridisation, purification or binding. Raw data were normalised using the geometric mean of robust housekeeping genes and volcano plots were used to identify the differential gene expression for each variable displays as a linear regression between the −log₁₀ of the p-value (using the Benjamini–Yekutieli adjustment) and log₂ fold change. Heat maps were created using hierarchical clustering (Euclidean distance as distance metrics and average as linkage method).

**PCR.** RNA, obtained from hippocampal tissue or isolated microglia (Nucleospin® RNAII KIT, Macherey-Nagel, Duren, Germany), was used to prepare cDNA (High-Capacity cDNA RT kit, Applied Biosystems, UK) for subsequent analysis of mRNA expression of CD68 (Mm03047343_m1) in hippocampus and Tyrobp (Mm00449152_m1), Cstd (Mm00515586_m1), Ccl6 (Mm01302419_m1), Trem2 (Mm04209424_g1), as well as the internal control, β-actin (Mm00407939_s1; Applied Biosystems, UK) in microglia. Gene expression was assessed using Applied Biosystems 7500 Fast Real-Time PCR machine and calculated relative to the endogenous control samples and to the control sample.

**Analysis of phagocytosis in isolated cells.** Aβ engulfment was assessed in microglia as previously described[8]. Briefly, microglia were isolated from WT and APP/PS1 mice, fixed in 4% PFA, washed in PBT (PBS + 1% Triton-X 100), incubated first, in PBT containing 3% bovine serum albumin (BSA; 1 h) to block

non-specific binding, and subsequently in the presence of rabbit anti-Iba1 (1:1000; Wako, Japan) and mouse anti-Aβ (1:500; Biolegend, US; 2 h, 4 °C). Coverslips were washed and incubated (2 h, room temperature) with Alexa Fluor 546 donkey anti-rabbit IgG (1:1000) and Alexa Fluor 488 donkey anti-mouse IgG (1:1000), and mounted in ProLong Gold with the nuclear marker DAPI (ThermoScientific, US). Images (8 fields/animal; ×40 magnification) were acquired with a Zeiss Axio Imager A1 microscope, analysed using ImageJ software, and calculated as % of Iba+ cells that engulfed Aβ.

**Metabolic analysis**

*SeaHorse technology.* Extracellular acidification rate (ECAR) was assessed using the glycolytic flux test (SeaHorse BioScience, US) as previously described[9]. Briefly, isolated microglia were washed according to the manufacturer's instructions, assay medium was added to give a final volume of 180 μl/well and the plate was incubated in a CO₂-free incubator (37 °C, 1 h) after which time glucose (10 mM), oligomycin (20 μM) and 2-deoxy-D-glucose (2-DG; 500 mM (all prepared in glycolytic flux assay media)) were sequentially delivered to the cells from the loaded ports at 24 min intervals. ECAR was automatically calculated using the SeaHorse XF96 software and 4–6 replicates were assessed for each separate sample.

*Metabolite analyses by mass spectrometry.* Microglia (200,000 cells) were washed three times with ice-cold PBS, plates were transferred on dry ice, ice-cold extraction buffer (250 μl in 80% LCMS grade methanol containing 0.05 ng/ml thymine-d4 and 0.10 ng/ml glycocholate-d4 as internal standards) was added. Samples were transferred to microtubes, vortexed and centrifuged (20,000×g, 10 min, 4 °C) and stored at −80 °C for later analysis. Extracted samples were subjected to LC/MS analysis as previously described[33]. Metabolite extracts were loaded onto a Luna-HILIC column (Phenomenex) using an UltiMate-3000 TPLRS LC with 10% mobile phase A (20 mM ammonium acetate and 20 mM ammonium hydroxide in 5:95 v/v acetonitrile/water) and 90% mobile phase B (10 mM ammonium hydroxide in 75:25 v/v acetonitrile/methanol). A 10 min linear gradient to 99% mobile phase A was used to separate metabolites. Subsequent analysis was carried out using a Q-ExactiveTM HF-X mass spectrometer (Thermo). Negative and positive ion modes were used with full scan analysis over m/z 70–750 m/z at 60,000 resolution, 1e6 AGC, and 100 ms maximum ion accumulation time. Additional MS settings were: ion spray voltage, 3.8 kV; capillary temperature, 350 °C; probe heater temperature, 320 °C; sheath gas, 50; auxiliary gas, 15; and S-lens RF level 40. Targeted processing of a subset of known metabolites and was conducted using TraceFinder software version 4.1 (ThermoFisher Scientific). Compound identities were confirmed using reference standards. In all cases, metabolite abundance was normalised using internal standards and relative changes were assessed.

**PFKFB3 subcellular localisation analysis.** Isolated microglia were cultured for 5 days, after which time they were fixed (4% PFA, 30 min), permeabilized (PBT (PBS + 0.3% Triton-X 100), 10 min), blocked (PBT + 3% BSA, 1 h) and incubated (4 °C, overnight) with goat anti-Iba1 (1:1000; LSBio Inc., US) and rabbit anti-PFKFB3 (1:250; Abcam, UK). Coverslips were washed and incubated (2 h, room temperature) with Cy5 Alexa Fluor 633 donkey anti-goat IgG (1:1000) and Alexa Fluor 488 donkey anti-rabbit IgG (1:1000) and mounted in ProLong Gold with the nuclear marker DAPI (ThermoScientific, US). Co-localisation of PFKFB3 and DAPI was assessed using confocal datasets with the Imaris software colocalization (coloc) module. Specifically, the number of PFKFB3 voxels colocalised with DAPI voxels was assessed and expressed as the percentage of total PFKFB3 voxels.

**Immunohistochemistry**

*Mouse samples.* Brain sections were used to assess amyloid pathology, PFKFB3, CD68 and Iba1. Sections were incubated with methanol (5 min, room temperature) and pepsin (20 min, room temperature) to facilitate antigen retrieval. Samples were washed, tissue was permeabilised (10 min; PBS with 0.3% Triton-X 100), blocked (1 h; 10% BSA with 0.3% Triton-X 100), incubated in primary antibodies (72 h, 4 °C; goat anti-Iba1 (1:1000; LSBio Inc., US), rabbit anti-PFKFB3 (1:250; Abcam, UK), mouse anti-Aβ (6E10,1:500; Biolegend, US), rat anti-CD68 (1:500, ThermoFisher Scientific)) washed, incubated in secondary antibodies (2 h, room temperature; donkey anti-goat IgG Cy5 (1:000, Abcam), Donkey anti-rabbit IgG Alexa Fluor® 488 (1:000, Abcam), Donkey anti-mouse IgG Alexa Fluor® 488 (1:000, Abcam), Donkey anti-rat IgG Alexa Fluor® 594 (1:000, Abcam)) and mounted in Pro-Long®Gold with DAPI (ThermoScientific, US). Amyloid plaques were detected using ThioS (1%, 10 min, room temperature) and cleared in ethanol (70%, 3×, 5 min).

*Human samples.* For DAB-immunoperoxidase staining, 10 μm-thick sections of parietal cortex from post-mortem tissue of male and female AD patients and age-matched controls were heated (37 °C, 3 h), deparaffinized in xylene (2 × 15 min, Sigma-Aldrich) and rehydrated by successive incubations in ethanol (100% × 10 min, 90% × 10 min, 75% × 10 min) and dH₂O (30 min). Sections were incubated in H₂O₂ solution (1% in 20% methanol, 20 min, room temperature) to quench endogenous peroxidase activity, rinsed in PBS, incubated with citrate-Na (pH 6.0) to facilitate antigen retrieval (2 × 5 min in the microwave) and washed in TBS (3 × 5 min). Sections were blocked in normal horse serum (20% in 0.2% Triton, 45

min), incubated with rabbit anti-Iba1 (1:1000; overnight, 4 °C in a humid tray chamber; Wako, Japan), washed with TBS, and incubated with biotinylated horse anti-rabbit secondary antibody (90 min), which was conjugated to HRP via the ABC method (Vector Laboratories) for a further 90 min before 3,3′-diamino-benzidine (DAB) was used as a chromogenic substrate. Sections were dehydrated in ascending ethanol concentrations and mounted on coverslips using DPX mounting media. Congo red staining was used to identify amyloid and, in this case, sections were incubated in an alkaline solution of 80% ethanol saturated with NaCl followed by a 0.2% Congo red solution.

*Image acquisition and analysis.* For DAB immune-peroxidase images, sections were cleared in xylene and imaged at ×40 magnification using an Olympus DRP72 camera mounted on an Olympus BX51 light microscope. For confocal immunofluorescence microscopy, sections were viewed using a Leica SP8 scanning confocal microscope and analysis of images was undertaken with ImageJ and Imaris software. ImageJ software (National Institute of Health, https://imagej.nih.gov/ij/) was used to convert images to 8-bit greyscale, a threshold was set and binarized images were filtered by pixel size to reduce background and enhance contrast. Cell perimeter, area, circularity and Feret's diameter (Particle Analysis, ImageJ) were measured[34].

In the case of mouse tissue, ramification (https://imagej.net/AnalyzeSkeleton) was also assessed and complexity was described by fractal dimension analysis[35]. Microglia were classified into five categories, Types I–V[36] including rod-shaped cells (Types IV and V) cells were counted and expressed as a % of the total number of Iba1+ cells per field. Confocal image stacks were converted to 3D images with the surface-rendering feature of Imaris BitPlane software (version 7.6.5). To assess Iba1 and CD68, confocal image stacks (Leica microscope, LACx programme) were analysed using ImageJ. Images were converted to 8 bits and masks and thresholds were applied to derive integrated density. For analysis of Aβ phagocytosis in vivo, Aβ encapsulated in CD68+ vesicles within Iba1+ microglia in mouse brain sections (×40 magnification) was assessed using confocal image stacks (BitPlane software, v 97.56.5) as described[37]. For analysis of amyloid-beta plaques and area, images were taken at ×40 magnification and converted to 8-bit binary images using ImageJ software. A threshold was set and images were filtered for pixel size to exclude artifacts and enhance contrast using the 'Analyse particles' tool. Congo red-positive plaques were selected using the region of interest (ROI) tool and the area of selection was measured. Mean values for plaque area (pixels$^2$) were calculated for each subject and data for group averages are presented.

**Statistics and reproducibility**. Data were analysed and figures prepared using GraphPad Prism (Version 9). Statistical analysis was carried out using Student's *t*-test for independent means or two-way ANOVA, followed by post hoc as appropriate, using Tukey's multiple comparisons test. The significance level was set at $p < 0.05$. Data are reported as the mean ± SEM and *n* refers to the number of separate replicates.

**Reporting summary**. Further information on research design is available in the Nature Research Reporting Summary linked to this article.

## Data availability
All raw data are available on request. The data from the Nanostring study will be deposited in an appropriate repository when publications are complete.

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

## Acknowledgements

This work was supported by Principal Investigator grants to M.A.L. from the Science Foundation Ireland (15/iA/3052 and 11PI/1014) for which we are very grateful. We wish to acknowledge Professor Stephen Gordon and Dr John Browne, UCD Conway Institute of Biomolecular and Biomedical Research, University College Dublin for their assistance with conducting the NanoString studies.

## Author contributions

A.R.A. and M.-V.G.-S.: metabolism, microscopy, and image analysis, V.M.: NanoString and image analysis, E.O.N., L.J., and A.S.G.: microscopy, cell morphology, and image analysis, E.L.M.: metabolomics, EC: supervision, M.L.: conceptualisation, project oversight, supervision, and writing. All authors: reviewing and editing.

## Competing interests

The authors declare no competing interests.

## Primary Handling Editor

Karli Montague-Cardoso. Peer reviewer reports are available.
