## [Peer Review File · Communications Biology]

Reviewers' comments:

Reviewer #1 (Remarks to the Author):

The authors investigated the role of sex-related differences of microglial cells in an Alzheimer's disease mouse model overexpressing mutant amyloid precursor protein (APP) and presenilin 1 (PS1). In detail, they analyzed changes in the expression of genes related with different states of microglial activation but also morphological, functional and metabolic alterations. Furthermore, the authors analyzed microglial sex-related morphological changes in the hippocampus and parietal cortex of post-mortem human brains of AD patients.

The topic of research is very interesting and novel, the results were well introduced and discussed, the used methods were the appropriate. However, in my opinion, there are some aspects of the manuscript that need to be clarified and some results need to be better or differently discussed. Some conclusions are not supported by the results.

UNDER Figure 1A: First, the image is essentially uninterpretable, making impossible to know which genes are down or over expressed in male or female APP/PS1 mice. I strongly suggest the authors to replace this figure with a better and higher quality one.

Furthermore, the authors analyzed 77 genes in isolated microglia from the brain of the different experimental groups and state that each one is related with a specific homeostatic function or it is characteristic for a concrete activation state of microglial cells. They also reported that most of them are altered in female APP/PS1 mice but they just mentioned a few genes in results and discussion paragraphs. Which other genes show an altered expression? In which function are they involved in microglial cells? Please, mention them in results and further discuss in which possible mechanism they are involved in AD.

Under Figure 2 results: graphs 2E, 2H and 2I suggest that microglial cells from WT male mice are more branched and have a bigger soma than microglia from female WT mice. These changes simply suggest that microglia from male mice in comparison with those from female mice are morphological different in physiological conditions and not that they are more activated as the authors state under lines 228-229.

In addition, cell perimeter, cell area and pixel/cell ratio of microglia in female APP mice (but not in male APP mice) are significantly reduced. These results suggest that in female APP mice, microglial cells efficiently respond to the environmental changes produced by the model. In contrast, microglia in male APP mice fail to respond (no changes are observed in comparison with male WT mice). These evidences disagree with the authors' conclusions stated under lines 232-234.

The authors' conclusion are instead supported by the morphological analysis performed in post-mortem human brains (Figure 7). There, it is very clear that microglia from female patients is unresponsive while microglia from male patients drastically change their morphology.

In order to avoid any misinterpretation of results, I strongly suggest the authors to differently discuss their results.

Concerning PFKFB3 results: It is not clear how the authors have quantified PFKFB3 staining in the nucleus and cytosol (by intensity?). Please detail it in methods. Furthermore, in order to better appreciate the changes due to the nuclear-cytosolic translocation, the authors should also show nucleus/cytosol ratio using raw data.

MINOR CONCERNS:

Scale bar value is missing in figures 5A and B and in figure 8.

Reviewer #2 (Remarks to the Author):

In the research article 'Microglial metabolism is a pivotal factor in sexual dimorphism in Alzheimer's disease', the authors show 1) preferential upregulation in genes linked with microglial activation in females compared to males, and 2) sex-related difference in microglial morphology, metabolism and function in the APP/PS1 mouse model contributing to the amyloid pathology, a

feature verified in human post-mortem tissue.

The authors conducted biochemical and histological evaluation in microglial cells, mouse and human samples with impressive amount of data. There are a couple of concerns that the authors should consider:

1. Whether the differential expression of microglial markers determine a distinctive morphology and the functional/metabolic state of the microglia. The authors do identify patterns that are selectively associated with female APP/PS1 mice compared to genotype-matched males; however, is there a change in the functional expression of the protein. It would help to further strengthen and reinforce the notion supporting the changes in microglial marker expression exhibited in female APP/PS1 mice.

2. Are rod-shaped microglia a feature associated with normal ageing? (PMID: 28131016 and PMID: 28522972). The reason for highlighting this point is the fact that the authors in the study utilized aged mice (16-18 months old) that may contribute to the appearance of rod-shaped microglia. It would be of great interest to know whether the rod-shaped microglia phenotype is a feature associated with the development of amyloid plaques, which in the APP/PS1 model is around 6-7 months old, and whether that is sex-related?

3. The authors specify quantification of "Types IV and V rod-shaped microglia, which are indicative of chronically-activated cells". It has been proposed the rod-shaped microglia may be a transitional state between ramified and amoeboid microglia (PMID: 28522972); thus, indicating an early feature which transforms into amoeboid microglia upon chronic insult. This may account for the increased plaque deposition observed in female APP/PS1 mice as well as human female post-mortem tissue.

4. How is the plaque load in the hippocampus of human post-mortem tissue? Figure 7 displays microglial morphology in human post-mortem tissue whilst the plaques shown in Figure 8 are labelled a cortical (parietal cortex). It would be ideal to have a direct association of plaque load and microglial morphology in a region-specific manner. Also, rod-shaped microglia express high levels of phagocytic markers (CD68/ED-1 and MHC-II), it would be of great interest to verify this in the human post-mortem tissue.

5. I would recommend structuring the heatmaps in a manner where the differential gene expression is grouped in clusters reflecting the morphological, functional, and metabolic characteristics of microglia. This would enable to dissect out the particular changes in microglia in a sex- and disease-specific manner. Also, in the text, specifying the full form of the gene and its association with microglial morphology/function will enable the flow of the text more smoothly.

Reviewer #3 (Remarks to the Author):

In this manuscript, the authors have well investigated sex- related differences through one of the markers being microglia in mouse model of AD (APP/PS1) as well as in post-mortem brain tissue from AD patients. They have shown changes in the associated genes that are indicative of microglial activation, those were increased in cells from female APP/PS1 mice. They have also shown distinction from males in terms of morphological, metabolically and functionally. They have shown shift in microglial metabolism towards glycolysis. They identified increased amyloid load in females compared with the male post-mortem samples from AD patients. Another marker of microglial activation, CD68 mRNA was shown higher in female mice compared to the male counterpart. Also, significant upregulation in glycolytic enzymes, Hk2, Pfkfb3, Gapdh, Pgk1 and Pgam1 have been shown by the authors that indicates shifting of metabolites that significantly greater in female mice compared to males.

This manuscript is well written with good representation of data.

The manuscript can be published.

Reviewer #1

The authors investigated the role of sex-related differences of microglial cells in an Alzheimer's disease mouse model overexpressing mutant amyloid precursor protein (APP) and presenilin 1 (PS1). In detail, they analyzed changes in the expression of genes related with different states of microglial activation but also morphological, functional and metabolic alterations. Furthermore, the authors analyzed microglial sex-related morphological changes in the hippocampus and parietal cortex of post-mortem human brains of AD patients.

The topic of research is very interesting and novel, the results were well introduced and discussed, the used methods were the appropriate. However, in my opinion, there are some aspects of the manuscript that need to be clarified and some results need to be better or differently discussed. Some conclusions are not supported by the results.

1. Figure 1A: First, the image is essentially uninterpretable, making impossible to know which genes are down or over expressed in male or female APP/PS1 mice. I strongly suggest the authors to replace this figure with a better and higher quality one. Furthermore, the authors analyzed 77 genes in isolated microglia from the brain of the different experimental groups and state that each one is related with a specific homeostatic function or it is characteristic for a concrete activation state of microglial cells. They also reported that most of them are altered in female APP/PS1 mice but they just mentioned a few genes in results and discussion paragraphs. Which other genes show an altered expression? In which function are they involved in microglial cells? Please, mention them in results and further discuss in which possible mechanism they are involved in AD.

RESPONSE:

We thank the reviewer for this suggestion. Figure 1A has been edited and is replaced with 3 heat maps that group genes as those described in the literature as homeostatic genes and genes that are upregulated in DAMs/ARMs and upregulated in injury/other inflammatory conditions like ALS. Figure S1 has been revised to group microglia according to these categories and an additional supplementary figure (Figure S2) is included that shows heat maps and histograms of genes associated with oxidative stress, inflammation, microglial motility/migration and antigen presentation. As suggested, the Results section has been edited to reflect on these changes and several sentences have been included in the Discussion highlighting genes that are altered in the context of microglial function.

The first paragraph of the results (lines 75-100) has been edited to read “Specifically, genes that define DAMs and/or ARMs (Figure 1A) ^{5,6} and genes that are upregulated in inflammatory conditions like ALS and injury (Figure 1B) were changed in a genotype- and sex-related manner and, specifically, were markedly increased in microglia from female APP/PS1 mice. These include genes that are linked with altered risk of AD like *ApoE*, *Bin1*, *Trem2* and *Cd33* ¹⁰. Sex-related differences in genes that identify homeostatic microglia were also observed with marked downregulation of *P2ry12* and upregulation of several genes that have been shown to increase in models of disease like *Ctsd*, *Fth1* and *Lyz2*, particularly in cells from female APP/PS1 mice (Figure 1C). Differentially-expressed genes are shown in the volcano plot (Figure 1D) and analysis of the mean data revealed that there was a significant genotype-related increase in genes that have been reported to characterise DAMs/ARMs like *Tyrobp*, *Ctsd*, *Ccl6* and *Trem2* (p* < 0.05; ***p* < 0.01; ****p* < 0.001; Figure 1E) and these changes were validated using RT-PCR (***p* < 0.01; ****p* < 0.001; Figure 1F). It is notable that changes in *Tyrobp*, *Ctsd* and *Ccl6* were significantly greater in microglia from APP/PS1 female mice compared with males ([§]*p* < 0.05; ^{§§}*p* < 0.01; Figure 1E). Similar genotype- and sex-related increases were observed on other genes characteristic of DAMs/ARMs including *Cst7*, *Aplp2*, *Cd74*, and *Axl*, and genes that are upregulated in injury and/or neuroinflammatory diseases other than AD including *Gpnmb* and *Plxdc2* (**p* < 0.05; **p* < 0.01; ****p* < 0.001 WT v APP/PS1; [§]*p* < 0.05; ^{§§}*p* < 0.01; ^{§§§}*p* < 0.001, male v female APP/PS1; Figure S1). There were no statistically significant changes in the 4 most-consistently described homeostatic genes, *P2ry12*, *Cx3cr1*, *Tmem119* or *Csfr1*, although genotype-related increases were observed in *Hexb*, *C1qa* and *C1qc* (**p* < 0.05; ***p* < 0.01; ****p* < 0.001) and sex-related differences in APP/PS2 mice in *Fth1* and *Lyz2* ([§]*p* < 0.05; ^{§§}*p* < 0.01; Figure S1). Genotype- and sex-related changes in genes that encode proteins associated with oxidative stress, inflammation, microglial migration/motility and antigen presentation were also observed and, in many cases, changes were more marked in microglia from female APP/PS1 mice (Figure S2).”**

The second paragraph of the Discussion (lines 223-237) has been edited and includes the following additional sentences “A number of the genes that exhibited sex-related differences were linked with

altered risk of AD, for example *Apoe*, *Bin1*, *Trem2* and *Cd33*¹⁰, as well as several members of the cathepsin family of cysteine proteinases. In addition, genes that are upregulated following injury and/or in models of diseases amyotrophic lateral sclerosis and multiple sclerosis^{1, 3}. Several genes indicative of oxidative stress and inflammation were also increased in female APP/PS1 mice compared with male APP/PS1 mice, including *Ubqin1*, *App* and *Hmox1* that play a role in regulating oxidative stress-induced signalling, and *Nkiras1*, which modulates NFκB signalling, and *Mef2a* and *Panx1* which regulate microglial surveillance and migration respectively. Marked sex-related differences were also observed in genes like *Itgam*, *Aif1* and *Vim*, which impact on microglial motility and *Atg3*, *Ctsl* and *Fcgr1*, which are involved in antigen presentation. The ramifications of such marked changes in gene expression may suggest that microglia from females are more primed for change than microglia from males and potentially more dysfunctional and less neuroprotective, perhaps explaining the sex-related difference in behaviour¹³.”

Changes to Figures (shown below)

Figure 1 has been edited so that the original heat map is replaced with 3 separate heat maps incorporating changes in genes that have been reported to be upregulated in DAMs/ARMs (A) injury/disease (B) and those that reflect the homeostatic state (C).

Figure S1 has been edited to show changes to genes in these 3 groupings.

Figure S2 is a new figure showing changes to genes that are associated with oxidative stress (A), inflammation (B) and 2 microglial functions, migration/motility (C) and antigen presentation (D).

Figure S1

Figure S2

2. Under Figure 2 results: graphs 2E, 2H and 2I suggest that microglial cells from WT male mice are more branched and have a bigger soma than microglia from female WT mice. These changes simply suggest that microglia from male mice in comparison with those from female mice are morphological different in physiological conditions and not that they are more activated as the authors state under lines 228-229.

3. In addition, cell perimeter, cell area and pixel/cell ratio of microglia in female APP mice (but not in male APP mice) are significantly reduced. These results suggest that in female APP mice, microglial cells efficiently respond to the environmental changes produced by the model. In contrast, microglia in male APP mice fail to respond (no changes are observed in comparison with male WT mice). These evidences disagree with the authors' conclusions stated under lines 232-234.

RESPONSE: We thank the reviewer for pointing out these issues. The paragraph in the Discussion relating to morphology has been edited (now 2 paragraphs) taking into account the comments of

this reviewer and reviewer 2. Specifically related to this reviewer's comments, the text has been edited to simply state that there are sex-related morphological differences in microglia in the absence of amyloid pathology (line 244-264). The sentence that conveyed a confused message has also been edited. The revised paragraphs read as follows (with the change pertinent to this comment underlined): *"Morphological features of microglia have often been used to describe activation state. Rod-shaped microglia are characteristic of age^{12, 15} as well as chronic disorders of the CNS including AD and subacute sclerosing panencephalitis, where they are considered to be a reflection of a persistently-activated cell¹¹. Thus one interpretation of the data is that the increased number of these cells in female APP/PS1 mice reflects a persistently-activated state that is not evident in males. However others have suggested that the cells represent a transitional state between ramified and amoeboid¹²; this interpretation may be consistent with the increased plaque deposition in females because the transformation to amoeboid is dependent on developing pathology. It is unclear whether the appearance of rod-shaped microglia precede amyloidosis although no genotype-related change was observed in hippocampal tissue prepared from 3 month-old APP/PS1 mice compared with WT mice (unpublished data).*

Microglia from male WT mice were more amoeboid than female WT mice, and consistent with this, soma size was greater and perimeter and cell area were decreased identifying sex-related differences in morphology in aged mice in the absence of amyloid pathology. The importance of this is that amoeboid cells are considered to be phagocytic and mobile^{16, 17, 18}. This is consistent with the proposal that upregulation of genes indicative of an activated/disease state exhibit reduced function. Analysis of morphology in microglia that were distal from plaques indicated a more amoeboid morphology in males perhaps consistent with increased phagocytosis and reduced pathology."

4. The authors' conclusion are instead supported by the morphological analysis performed in post-mortem human brains (Figure 7). There, it is very clear that microglia from female patients is unresponsive while microglia from male patients drastically change their morphology. In order to avoid any misinterpretation of results, I strongly suggest the authors to differently discuss their results.

RESPONSE: We thank the reviewer for this comment. The Discussion has been edited to reflect this. The text (lines 309-320) now reads as follows, with the key changes underlined: *"Furthermore, microglia from male AD patients were amoeboid with little evidence of heterogeneity in morphology and this contrasted sharply with brains from female AD patients where there was marked heterogeneity with few amoeboid cells, some ramified cells, and numerous rod-shaped microglia, which were also observed in greater numbers in sections from female APP/PS1 mice. However, in contrast to the findings in WT mice, there were few amoeboid microglia in tissue from age-matched control males.*

The parallel increases in amoeboid microglia and increased CD68 immunoreactivity in microglia from male AD patients is consistent with previous data that suggest amoeboid cells are phagocytic¹⁸. One possible interpretation of the findings is that there may be a greater preservation of function, particularly responsiveness to pathological stimuli, in microglia from male, compared with female, AD patients."

5. Concerning PFKFB3 results: It is not clear how the authors have quantified PFKFB3 staining in the nucleus and cytosol (by intensity?). Please detail it in methods.

RESPONSE: The Methods now include a statement relating describing the protocol. It reads (lines 447-449) as follows: *'Specifically the number of PFKFB3 voxels colocalised with DAPI voxels was assessed and expressed as the percentage of total PFKFB3 voxels.'*

6. Furthermore, in order to better appreciate the changes due to the nuclear-cytosolic translocation, the authors should also show nucleus/cytosol ratio using raw data.

RESPONSE: An additional panel showing the requested data has been included in Figure 5D (as shown below). The Results section makes reference to this inclusion with a modified sentence, with the change (that appears at lines 162-163 in the text); the revised sentence, with the change underlined below, reads as follows: *"However PFKFB3 is activated only when it translocates from the nucleus to the cytosol and cytosolic PFKFB3 staining was significantly increased in cells from female APP/PS1 mice compared with males (^{§§} $p < 0.01$; Figure 5C-E) and this was accompanied by a significant decrease in nuclear staining (^{§§} $p < 0.01$) and a significant reduction in nuclear:cytosolic PFKFB3 ([§] $p < 0.05$) indicating preferential enzyme activation, consistent with the glycolytic signature, in microglia from female APP/PS1 mice."*

MINOR CONCERNS:

Scale bar value is missing in figures 5A and B and in figure 8.

RESPONSE: Scale bars have been included in all figures.

Reviewer #2

In the research article ‘Microglial metabolism is a pivotal factor in sexual dimorphism in Alzheimer’s disease’, the authors show 1) preferential upregulation in genes linked with microglial activation in females compared to males, and 2) sex-related difference in microglial morphology, metabolism and function in the APP/PS1 mouse model contributing to the amyloid pathology, a feature verified in human post-mortem tissue.

The authors conducted biochemical and histological evaluation in microglial cells, mouse and human samples with impressive amount of data. There are a couple of concerns that the authors should consider:

1. Whether the differential expression of microglial markers determine a distinctive morphology and the functional/metabolic state of the microglia. The authors do identify patterns that are selectively associated with female APP/PS1 mice compared to genotype-matched males; however, is there a change in the functional expression of the protein. It would help to further strengthen and reinforce the notion supporting the changes in microglial marker expression exhibited in female APP/PS1 mice.

RESPONSE: We thank the reviewer for this comment and have added an additional supplementary figure (Figure S6, shown below) to address the comment. This figure shows an increase in CD68 immunoreactivity in Iba1⁺ cells that is more marked in sections from male APP/PS1 compared with male WT mice and also more marked compared with female APP/PS1

mice, possibly reflecting increased phagocytosis. To check this, we assessed uptake of A β and fluorescently-labelled beads in microglia from neonatal mice that were cultured for 7 days and show that amoeboid Iba1⁺ microglia engulfed more A β and beads than microglia with processes. We also assessed CD68 as a proxy marker of phagocytosis in microglia isolated from adult mice and show that CD68 staining was more marked in amoeboid Iba1⁺ microglia compared with rod-shaped microglia. Reference is made to this in the Results section (lines 171-174) and the revised sentence reads as follows: *“Interestingly the relatively greater functionality in microglia from male mice is also suggested by the characteristic amoeboid morphology and supported by increased CD68 immunoreactivity in Iba1⁺ cells, increased phagocytosis of A β and fluorescently-labelled beads in amoeboid, compared with non-amoeboid microglia, and also increased CD68 staining in amoeboid microglia (Figure S6)’.*

We also assessed CD68 in sections from cortex of AD patients and age-matched controls and have included a new panel in Figure 8 (shown below) and added a sentence to the Results (lines 201-207) which reads as follows: *“Because of our findings in mice, we used CD68 as a proxy marker of phagocytic microglia and asked if CD68 immunoreactivity was more pronounced in the parietal cortex from male, compared with female, AD patients. The data revealed an increase in CD68 staining in parietal cortex of male, compared with female, AD patients (*p < 0.05; 1-tailed t test; Figure 8C). The pattern of staining was different in male and female AD patients with apparently greater clustering of CD68 in sections from female patients.”*

The penultimate paragraph of the Discussion (lines 316-320) has been edited to make reference to these findings and reads as follows: *“The parallel increases in amoeboid microglia and increased CD68 immunoreactivity in microglia from male AD patients is consistent with previous data that suggest amoeboid cells are phagocytic¹⁸. One possible interpretation of the findings is that there may be a greater preservation of function, particularly responsiveness to pathological stimuli, in microglia from male, compared with female, AD patients. In this context, a recent meta-analysis revealed that, when all variables were accounted for, the cognitive abilities of women with AD patients were worse than men with AD³¹.”*

Figure S6

Figure 8

2. Are rod-shaped microglia a feature associated with normal ageing? (PMID: 28131016 and PMID: 28522972). The reason for highlighting this point is the fact that the authors in the study utilized aged mice (16-18 months old) that may contribute to the appearance of rod-shaped microglia. It would be of great interest to know whether the rod-shaped microglia phenotype is a feature associated with the development of amyloid plaques, which in the APP/PS1 model is around 6-7 months old, and whether that is sex-related?

RESPONSE: We agree that a comprehensive analysis of age-related changes in rod-shaped microglia would be of great interest. However we assessed rod-shaped microglia in hippocampus only from 4 month-old WT and APP/PS1 mice in preliminary experiments and found that there was no genotype-related difference as indicated below; no sex-related difference was observed but numbers were small. The Results makes reference to this point stating (lines 111-114) “We found no genotype-related difference in rod-shaped microglia in hippocampus of 3-4 month-old mice (8.72 ± 1.15 and 12.35 ± 1.32 in WT ($n=6$, 3m, 3f) and APP/PS1 ($n=5$, 3m, 2f) mice respectively; mean \pm SEM; % of total microglia).”

3. The authors specify quantification of “Types IV and V rod-shaped microglia, which are indicative of chronically-activated cells”. It has been proposed the rod-shaped microglia may be a transitional state between ramified and amoeboid microglia (PMID: 28522972); thus, indicating an early feature which transforms into amoeboid microglia upon chronic insult. This may account for the increased plaque deposition observed in female APP/PS1 mice as well as human female post-mortem tissue.

RESPONSE: We thank the reviewer for making this point which is now included point in the Results and also in the discussion. The Results section has been edited (lines 107-108) to read as follows: “We classified microglia according to Types I-V (Figure S2) and quantified Types IV and V rod-shaped microglia, which are considered to be indicative of chronically-activated cells¹¹ or a transitional state between ramified and amoeboid¹².” The Discussion has been edited (lines 244-264) to read as follows: “Morphological features of microglia have often been used to describe activation state. Rod-shaped microglia are characteristic of age^{12, 15} as well as chronic disorders of the CNS including AD and subacute sclerosing panencephalitis, where they are considered to be a reflection of a persistently-activated cell¹¹. Thus one interpretation of the data is that the increased number of these cells in female APP/PS1 mice reflects a persistently-activated state that is not evident in males. However others have suggested that the cells represent a transitional state between ramified and amoeboid¹²; this interpretation may be consistent with the increased plaque deposition in females because the transformation to amoeboid is dependent on developing pathology. It is unclear whether the appearance of rod-shaped microglia precede amyloidosis although no genotype-related change was observed in hippocampal tissue prepared from 3 month-old APP/PS1 mice compared with WT mice.”

4. (i) How is the plaque load in the hippocampus of human post-mortem tissue? (ii) Figure 7 displays

microglial morphology in human post-mortem tissue whilst the plaques shown in Figure 8 are labelled a cortical (parietal cortex). It would be ideal to have a direct association of plaque load and microglial morphology in a region-specific manner. (iii) Also, rod-shaped microglia express high levels of phagocytic markers (CD68/ED-1 and MHC-II), it would be of great interest to verify this in the human post-mortem tissue.

RESPONSE: (i) We have added a statement indicating how plaque load is assessed (lines 500-505) “For analysis of amyloid beta plaques and area, images were taken at 40x magnification and converted to 8-bit binary images using ImageJ software. A threshold was set and images were filtered for pixel size to exclude artifacts and enhance contrast using the “Analyse particles” tool. Congo red-positive plaques were selected using the region of interest (ROI) tool and the area of selection was measured. Mean values for plaque area (pixels²) were calculated for each subject and data for group averages are presented.”

(ii) We agree with the reviewer that data from the same area should be presented and have therefore undertaken additional experiments in parietal cortex where we had some remaining tissue. We assessed morphological changes and these data are now presented in a revised Figure 7.

(iii) As suggested by the reviewer, we analysed CD68 by immunohistochemistry and the data are included in the revised Figure 8 (shown above in response to point 1). The findings indicate that there is an increase in CD68 staining in parietal cortex of male AD patients but not females but CD68 does not identify cell morphology. However, when this finding is considered with the data presented in Figure S6 (showing increased phagocytosis in amoeboid cells and increased CD68 in amoeboid microglia in mice), it is consistent with the idea that the amoeboid morphology of microglia in males reflect a greater phagocytic capacity compared with the mixed morphology that is evident in microglia from female AD patients. This point is made in the Results (lines 201-207) as indicated in the response to this reviewer’s first point, which also touched in this issue.

5. I would recommend structuring the heatmaps in a manner where the differential gene expression in grouped in clusters reflecting the morphological, functional, and metabolic characteristics of microglia. This would enable to dissect out the particular changes in microglia in a sex- and disease-specific manner. Also, in the text, specifying the full form of the gene and its association with microglial morphology/function will enable the flow of the text more smoothly.

RESPONSE: We are grateful to the reviewer for this suggestion, which was also made by Reviewer 1 (point 1). As indicated in our response to this point, which is on page 1 of these responses, Figure 1 has been revised and a new supplemental figure (Figure S2) has been added. The text has been edited accordingly.

Reviewer #3

In this manuscript, the authors have well investigated sex- related differences through one of the markers being microglia in mouse model of AD (APP/PS1) as well as in post-mortem brain tissue from AD patients. They have shown changes in the associated genes that are indicative of microglial activation, those were increased in cells from female APP/PS1 mice. They have also shown distinction from males in terms of morphological, metabolically and functionally. They have shown shift in microglial metabolism towards glycolysis. They identified increased amyloid load in females compared with the male post-mortem samples from AD patients. Another marker of microglial activation, CD68 mRNA was shown higher in female mice compared to the male counterpart. Also, significant upregulation in glycolytic enzymes, Hk2, Pfkfb3, Gapdh, Pgk1 and Pgam1 have been shown by the authors that indicates shifting of metabolites that significantly greater in female mice compared to males.

This manuscript is well written with good representation of data. The manuscript can be published.

RESPONSE: We thank the reviewer for these positive comments.

REVIEWERS' COMMENTS:

Reviewer #1 (Remarks to the Author):

The authors have solved all my concerns and the manuscript is much improved.

Reviewer #2 (Remarks to the Author):

This revised manuscript is well written and has addressed the revisions appropriately. The manuscript can be published.

Reviewer #3 (Remarks to the Author):

I do not have any comments on the reviewed manuscript.